# Future Regional Contributions for Climate Change Mitigation: Insights from Energy Investment Gap and Policy Cost

**Hongjie Sun [1,2,3],\* , Shuwen Niu [1],\* and Xiqiang Wang [4]**

1   College of Earth and Environmental Sciences, Lanzhou University, Lanzhou 730000, China
2   School of Economics and Management, Lanzhou University of Technology, Lanzhou 730050, China
3   School of Economics and Management, Southeast University, Nanjing 211189, China
4   Qilian Alpine Ecology and Hydrology Research Station, Key Laboratory of Ecohydrology of Inland River
    Basin, Northwest Institute of Eco-Environment and Resources, Chinese Academy of Sciences,
    Lanzhou 730000, China; wangxq@lzb.ac.cn
\*   Correspondence: sunhj15@lzu.edu.cn (H.S.); shuwenn@lzu.edu.cn (S.N.)

**Abstract:** Mitigating climate change and ensuring regional equity development is equitable are matters of global concern. Systematic and in-depth research into these issues is seldom conducted. In this research we combine qualitative and quantitative studies and use six state-of-the-art energy-economy analysis models and four long term scenarios to explore the distribution of regional contributions for climate change mitigation in the future. We focus on the energy investment gap and policy cost. The study's conclusion is that, under the assumption of carbon tax as a source of energy investment from 2025, the global positive energy investment gap in the climate change mitigation scenario will not appear until around 2035–2040. Asia and OECD90+EU (Countries from the OECD 1990, EU and its candidates) are the regions that will have a significant direct impact on the global energy investment gap under climate policies in the future. However, from the perspective of the relative value (the percentage of the energy investment gap relative to the energy investment in the Current Policies (CPol) scenario), Asia will contribute the most to the global energy investment gap under the climate stability policies. Under the Nationally Determined Contributions (NDC) scenario, Asia will contribute the most in the near term and REF will contribute the most in the medium term. The findings show that OECD90+EU will bear more cost in the pledges scenario, and Asia will bear more cost in the climate stability scenarios in the medium term. Contrary to the common sense expectation, the developed regions will contribute the least in terms of the proportion of the policy cost to the respective economic aggregates under the climate stability scenarios in the medium and long term, but the opposite is true in the developing regions. By and large, from the perspective of the current climate change mitigation policies, the developed regions and developing regions will achieve a win-win situation in the long run, but the relative contribution of the developed regions is not as great as was previously expected. These novel findings should prove to be useful to policy makers when developing transition strategies for climate change mitigation.

**Keywords:** energy investment; policy cost; energy mix; regional contribution; climate policy

---

## 1. Introduction

Climate change mitigation and equitable development are among the most important and complex challenges that humanity is facing today [1–3]. A large body of scientific literature shows that climate change mitigation can only be achieved through cooperation among nations due to the global negative externality of greenhouse gas emissions [4–6]. Several international agreements have been established to

address this global climate change mitigation cooperation; these include the United Nations Framework Convention on Climate Change (UNFCCC) [7], the Kyoto Protocol [8] and its Doha Amendment [9], the Copenhagen Accord [10], the Cancun Agreements [11], and the Paris Agreement [12]. Although these international agreements have stated action measures and targets for climate change mitigation, some crucial issues remain unresolved, particularly the question of under what conditions and when, developed regions need to provide financial assistance to developing regions, and what changes in their respective policy costs will occur. Furthermore, what are the differences in regional contributions to climate change mitigation?

Carbon tax, which is a levy on fossil fuels that produce carbon dioxide and can be paid either by the producers or consumers of fossil fuels, has always been seen as an important policy tool for controlling carbon emissions and for climate change mitigation [13]. The implications of carbon tax for the energy investment gap needed to meet the different goals of climate change mitigation are poorly understood. Previous studies have analyzed investments in a single country or region; these included Asia [14], Malaysia [15], and Latin America [16]. At the same time, it is generally believed that the sources of energy investment for climate change mitigation includecarbon tax revenue, private capital [17], and government capital [18]. Except for the Hyder paper [19], there has been little research proposing to recycle carbon tax revenue to fund an integrated investment in energy for climate change mitigation. The existing studies agree on the significant investment gap between climate policy and the "business as usual" projections [19], but due to limitations on the availability of the region-level energy investment and carbon price across multiple scenarios, no previous analysis has considered the impact of a carbon tax on the future energy investment gap in various regions. The policy costs are also central to understanding and implementing climate change mitigation policies. There is a broad body of literature on the policy costs caused by climate change mitigation. Most of the existing research has used single data to look into the policy cost at the global [14–16,19,20] or national levels [13,21,22]. Although some studies have explored the factors affecting the reduction of the policy cost [14,17,21], they have typically lacked a systematic comparative analysis. Few studies have focused on work sharing in a multi-model integration environment and assessed the contribution to climate change mitigation across the major economies.

In this study, we carry out a systematic quantitative analysis (multi-model with different functional structures) and address three questions to fill this gap. First, how would climate policies change the energy system compared with a business as usual projection? Second, what is the distribution of the energy investment gap under the premise of carbon tax as the source of the energy investment? Finally, what would be the regional contribution to the global policy costs, especially the comparison between the climate policy scenarios? Previous studies have assessed these issues independently, whereas this study uses a consistent multi-model integration approach to analyze these issues. Although these do not capture all the information related to climate change mitigation decision-making, this analysis provides useful insights into the regional impacts of climate policy and regional strategic incentives. Additionally, it allows us to discern the potential regional winners and losers from climate policies.

## 2. Methodology and Data

Based on the CD-LINKS (Linking Climate and Development Policies—Leveraging International Networks and Knowledge Sharing) project [23], this study employed six different integrated models to investigate the future contribution of five regions worldwide for climate change mitigation in four possible scenarios. The basic idea is to explore the complex interplay of the relationships among multiple variables (including climate change, energy structure, energy investment, carbon tax, and policy cost) from both the global and regional perspectives and to simulate the extent of the impact of people's efforts on climate change mitigation under different policy scenarios. The information provided from this process will provide the information needed to design complementary climate-development policies. The base year for the calibration is 2015, the value of all currencies is in constant 2015 USD, and the market exchange rates are used to convert the national currencies.

*2.1. Models in This Study*

This study is based on modeling energy-economy-climate systems and employs six models. These integrated assessment models (IAMs) each have multiple variables and a complex structure, but their modeling frameworks and solution algorithms are different. Due to the differences and complementarities between these models, they play an important role in the relevant scientific debates [19]. The analyses apply six models to assess the future regional contributions for climate change mitigation, while the details and specific strengths of each model are preserved. A brief description of the six models is as follows:

(1) AIM/CGE (Asia-Pacific Integrated Modeling/Computable General Equilibrium Model) is a recursive dynamic general equilibrium model covering almost all regions of the world. The model structure and mathematical formulas are detailed in [24]. In order to properly assess the bio-energy and land use competition, the agricultural sector is also highly decomposed [25].

(2) IMAGE (Integrated Model to Assess the Global Environment) is a comprehensive framework of the ecological environment model that simulates the environmental consequences of global human activities. The IMAGE framework uses exogenous assumptions about economic development, population, lifestyle, technological changes, and policies [26].

(3) MESSAGEix-GLOBIOM (Model for Energy SupplyStrategy Alternatives and their General Environmental Impact-Global Biosphere Management Model) integrates an energy engineering model and land use model into a global integrated assessment modeling framework through soft links [27,28]. The "ix" platform is used for comprehensive and cross-sectoral modeling [29].

(4) The POLES (Prospective Outlook on Long-term Energy Systems) model is a partial equilibrium simulation model for the energy sector that covers the globe and has annual steps [30].

(5) REMIND-MAgPIE (Regional Model of Investment and Development and Model of Agricultural Production and its Impacts on the Environment) is a framework consisting of an energy economy-climate model and land use model. REMIND is a general equilibrium model of the energy economy that links the macroeconomic growth model with the energy system model [31]. MAgPIE is a global multi-regional economic land use optimization model that aims to analyze the situation before 2100 [32]. This is a partial equilibrium model of the agricultural sector that is solved in the recursive dynamic mode.

(6) WITCH (World Induced Technical Change Hybrid) is a comprehensive assessment model that is mainly used to assess the policies of climate change mitigation and adaptation. The multi-region and cross-temporal dimensions of this model help it to assess the best responses to several climate and energy policies across regions and time. A detailed description of the model can be found in [33,34].

*2.2. Scenarios*

Scenarios are widely used to explore the impacts of possible future climate change as a result of human activities and to assess the mitigation potential by modeling the energy system responses to mitigation efforts [35]. The goal of multiple scenarios is not to accurately predict the future, but to better understand uncertainty so that robust decisions can be made in all possible future scenarios [36,37].

Four scenarios were simulated in this study, which are summarized in Table 1. Current Policies (CPol) considers the high-impact energy and climate-related policies implemented in the G20 countries as of 2015. Nationally Determined Contributions (NDC) assumes the implementation of all countries' NDCs (conditional commitments) by 2030, which is the target year for most of these. "2 °C" expresses the aim to hold the maximum increase in global average temperatures to 2 °C (above the pre-industrial level) over the course of the 21st century with >66% likelihood. "1.5 °C" denotes the aim to limit the increase in global average temperatures to 1.5 °C (above the pre-industrial level) in 2100 with >50% likelihood. Moreover, we chose the supply-side carbon tax here because the energy investment is targeted at the supply-side energy. There are significant differences in the preconditions for setting these policy scenarios (Table 1). This has led to a series of differences in the energy structure, $CO_2$ emissions, energy investment gap, and climate change mitigation range under different scenarios. Four

scenarios were used as a basis to explore the synergies and trade-offs between sustainable development and climate change. These policies for the impact of each scenario on each region are beyond the scope of this manuscript; they need further research and exploration.

**Table 1.** Overview of four scenarios.

| Scenario Names | Original Names | Description |
|---|---|---|
| CPol | NPi | National Policies until 2030, No Policy after 2030 |
| NDC | INDCi | National Policies until 2020, followed by the implementation of NDCs (Nationally Determined Contributions) until 2025/2030 |
| 2 °C | NPi2020_1000 | National Policies until 2020, will keep the budget of 2011–2100 within 1000 GtCO$_2$ by 2020, and the probability of staying below 2 °C in the 21st century is >66% |
| 1.5 °C | NPi2020_400 | National Policies until 2020, will keep the budget of 2011–2100 within 400 GtCO$_2$ by 2020, and the probability of staying below 1.5 °C in the 21st century is >50% |

This table shows the names of the scenarios and indicates the links with those used in the CD-LINKS project. Source: Modified from the CD-LINKS Scenario Database [38].

All the data for the scenarios are available in [38]. Meanwhile, we follow the previous literature on the population and GDP (Gross Domestic Product) assumptions under Shared Socioeconomic Pathway 2 [39]. It should be noted that the scope of this study is not only global but also regional; Table 2 provides the details. According to the regional classification in the CD-LINKS scenario database, we discuss the issue with regard to five regions. In view of the limitations of the CD-LINKS database, relevant research at the national and sub-national level is not covered here.

**Table 2.** Overview of five regions.

| Regional Acronyms | Description (Regional Definition) |
|---|---|
| OECD90+EU | Countries from the OECD 1990, EU and its candidates. |
| REF | Countries from the Former Soviet Union. |
| ASIA | Countries from Asia, except the Former Soviet Union as well as the Middle East and Japan. |
| MAF | Countries from the Middle East and Africa. |
| LAM | Countries from Latin America and the Caribbean. |

Source: Modified from CD-LINKS Scenario Database [38].

## 3. Results and Discussion

### 3.1. Effect of Climate Change Mitigation Policies on Energy Mix

#### 3.1.1. Primary Energy

Primary energy is a form of energy generated in nature without any human processing [40,41]. Fossil fuels currently dominate the primary energy supply. The largest source of greenhouse gas produced by human activities is from burning fossil fuels. Therefore, the policies to limit emissions will affect the amount and structure of primary energy. This is illustrated in Figure 1 and Table 3, which depict the trajectory of the global primary energy transition under different policy scenarios (2015–2100).

A stricter climate change mitigation policy is associated with a higher carbon price, which will limit the consumption of primary energy. In the short to medium term, the high carbon price will lead to a decrease in the global primary energy supply by 6.81% (6.90%) (to express data from different years we use parentheses), 19.46% (29.55%), 26.15% (35.16%), compared with the baseline (business as usual) projection in the NDC, 2 °C and 1.5 °C scenarios, respectively, in 2030 (2050).

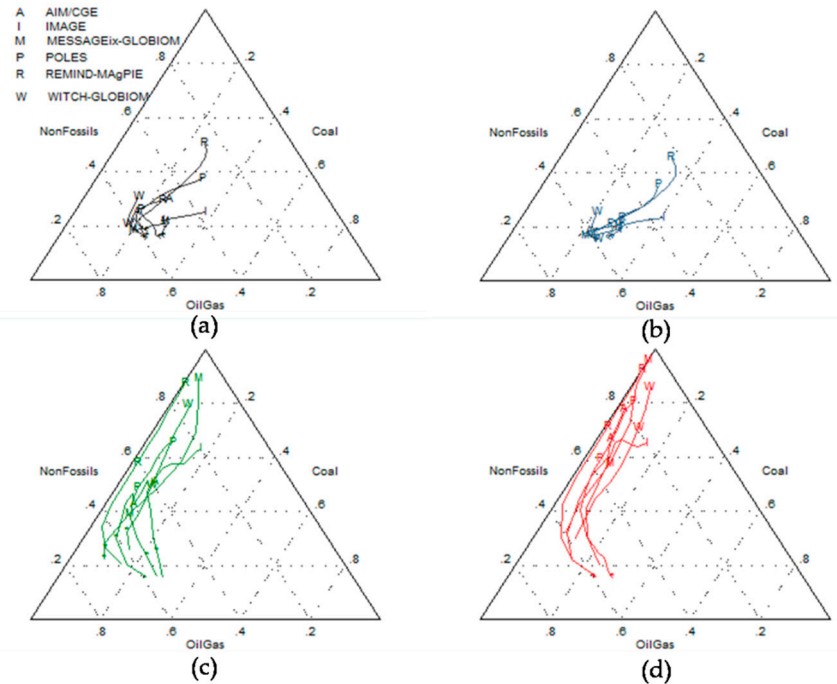

**Figure 1.** Trajectory of global primary energy transformation (2015–2100). (**a**) Current Policies (CPol) scenario (grey); (**b**) Nationally Determined Contributions (NDC) scenario (navy); (**c**) 2 °C scenario (green); (**d**) 1.5 °C scenario (red). Note that * indicates the starting point for 2015; capital letters represent mid-2050 and forward to 2100 corresponding to different models.

**Table 3.** The change rate of the global primary energy supply for the climate policy scenarios (NDC, 2 °C, and 1.5 °C), as opposed to the business as usual scenario, expressed as percentages.

| Scenarios | 2015 | 2020 | 2025 | 2030 | 2035 | 2040 | 2045 | 2050 |
|---|---|---|---|---|---|---|---|---|
| **NDC Scenario** | | | | | | | | |
| Coal | 0 | −3 | −11 | −20 | −24 | −24 | −27 | −26 |
| Gas | −1 | −1 | −5 | −7 | −8 | −7 | −8 | −7 |
| Oil | 0 | 0 | −3 | −6 | −5 | −5 | −5 | −4 |
| Nuclear | 1 | 3 | 13 | 18 | 23 | 26 | 26 | 28 |
| Biomass | −1 | 0 | 2 | 6 | 4 | 5 | 6 | 7 |
| Non-biomass renewables | 0 | 1 | 9 | 15 | 20 | 20 | 24 | 20 |
| Total primary energy | 0 | −1 | −4 | −7 | −8 | −7 | −8 | −7 |
| **2 °C Scenario** | | | | | | | | |
| Coal | 0 | −1 | −29 | −55 | −70 | −78 | −82 | −81 |
| Gas | 0 | 0 | −12 | −21 | −37 | −36 | −47 | −44 |
| Oil | 0 | 0 | −6 | −13 | −25 | −30 | −36 | −41 |
| Nuclear | 1 | 2 | 22 | 44 | 51 | 67 | 58 | 92 |
| Biomass | 0 | 0 | 2 | 6 | 21 | 43 | 65 | 84 |
| Non-biomass renewables | 0 | 1 | 13 | 33 | 63 | 65 | 78 | 73 |
| Total primary energy | 0 | 0 | −11 | −19 | −28 | −29 | −31 | −30 |
| **1.5 °C Scenario** | | | | | | | | |
| Coal | 0 | −2 | −41 | −66 | −79 | −83 | −86 | −86 |
| Gas | 0 | 0 | −19 | −31 | −48 | −48 | −61 | −59 |
| Oil | 0 | 0 | −11 | −25 | −36 | −50 | −58 | −69 |
| Nuclear | 1 | 4 | 32 | 65 | 69 | 96 | 73 | 138 |
| Biomass | −2 | 0 | 4 | 20 | 34 | 69 | 106 | 140 |
| Non-biomass renewables | 0 | 1 | 18 | 45 | 74 | 85 | 102 | 101 |
| Total primary energy | 0 | 0 | −17 | −26 | −35 | −35 | −37 | −35 |

From the perspective of the primary energy mix, in the medium and long term, coal shows an uptrend under the CPol and NDC scenarios, and a downtrend in the strict climate change mitigation policy scenarios. Oil and gas show a stable preservation in the CPol and NDC scenarios, and a decline in the strict climate change mitigation policy scenarios. Non-fossil energy supply shows an upward trend across all scenarios (Figure 2). Taking the 2 °C scenario as an example, the global carbon supply (model mean) will decline by 47% (70%), the gas supply will increase by 10% (3%), and oil supply will increase by 2% (−23%), while solar energy will increase tenfold (47-fold) and wind energy will increase fivefold (15-fold) between 2015 and 2030 (2050).

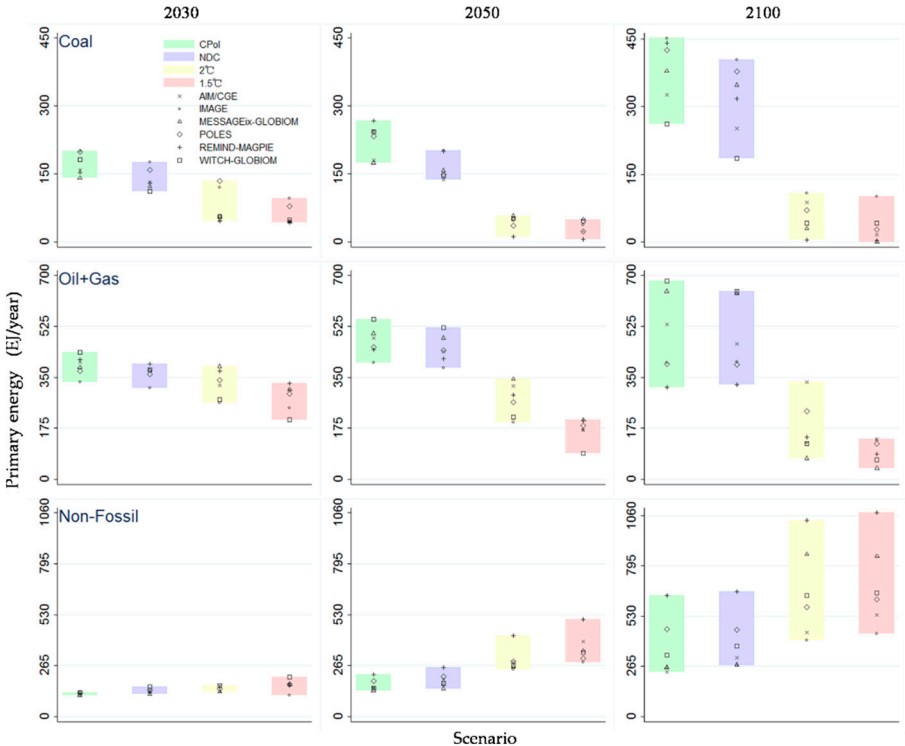

**Figure 2.** Primary energy development across models under CPol and climate policy scenarios (EJ/year). The colors represent scenarios, and the symbols represent models.

Table 3 decomposes the change in the primary energy supply in the climate change mitigation scenarios. It further proves that the supply of fossil energy decreases year by year and the proportion of non-fossil energy increases from 2025 onwards, which is consistent with the results presented by [42]. The impact of NDC on the primary energy supply, especially global oil, natural gas, and biomass, is not significant compared with the stringent policy scenarios (Table 3 and Figure 2). This is due to inefficiencies caused by the fact that some fossil fuel rich countries have not joined the Nationally Determined Contributions (NDCs) or fulfilled their commitments [42,43].

The projected regional primary energy is unequally distributed among the regions due to differences in the scale and patterns of consumption. The projected regional results vary as exhibited in Figures S1 and S2.

### 3.1.2. Electricity Generation

This part mainly analyzes the energy mix of power production under the scenarios in 2030 and 2050. From a global perspective, under the policy scenarios of climate change mitigation, power production will decline to a certain extent in the short term (until 2030) and will grow from the medium to long run, compared to the CPol scenario (Figure 3). This mainly depends on the changes in the carbon price, technology composition and technology cost. In other words, the rising carbon price

brought by climate policy restrictions leads to a decline in fossil fuels consumption, such as coal in power generation, while the increase in the non-fossil energy types of power generation cannot maintain the power production in the short run. With the perfection and popularization of low-carbon energy technology, the descending cost of low-carbon energy leads to the increase in power production in the long run. This result is inconsistent with [42]. Simultaneously, it is worth noting that the fossil fuels share accounts for a small proportion of electricity generation and can be neglected in 2050 under the stringent scenarios. Renewable energy would become a very important factor in global mitigation efforts, especially the leading role of solar and wind energy in power generation around 2050. We can see from Figure 3 that the share of clean power (including hydropower, wind, solar, and nuclear power) will increase significantly under the stricter climate policy.

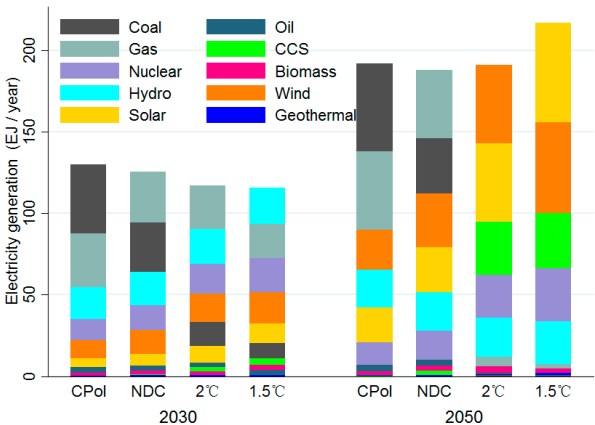

**Figure 3.** Electricity generation by technology in different scenarios in 2030 and 2050 at the global level (model mean).

The analyses of the global level discussed above conceal regional differences. The following breaks down these results in order to better understand the changes in power production technologies between different regions under different scenarios. From a regional perspective, the variation trend of energy types used in power production is similar to the global level, but there are significant differences between the regions across the scenarios as defined for CPol, NDC, 2 °C, and 1.5 °C (Figure S3).

Gas is the dominant source of electricity generation in most regions under both CPol and NDC scenarios, with the exception of Asia and LAM, where coal is the largest power source in Asia and hydropower is the largest power source in LAM. It is highlighted here that the carbon share in power generation in Asia affects the corresponding global level under the CPol scenario. Under the scenarios of stringent climate change mitigation policies (2 °C, 1.5 °C), in the short term (2030), the energy used for global power generation is diversified and balanced, but the regional differences are obvious. Renewable energy is the dominant technology for power generation under the stringent scenarios (2 °C, 1.5 °C) across all regions in the long run (2050).

Figure 4 sheds light on unit operating costs of various power generation technologies in the 2 °C scenario during 2015–2050 (other scenarios' curves have a similar trend). Identifying the relationship between the operating cost of power generation technologies and the installed capacity has important implications for finding the optimal emission reduction paths. As pointed out by [42], the wider use of renewable energy is accelerated by climate change mitigation actions. Figure 4 shows that there is an inverse relationship between the cumulative capacity and the operating costs in power generation technologies. This means that the expansion of installed capacity is favorable for operating costs reductions.

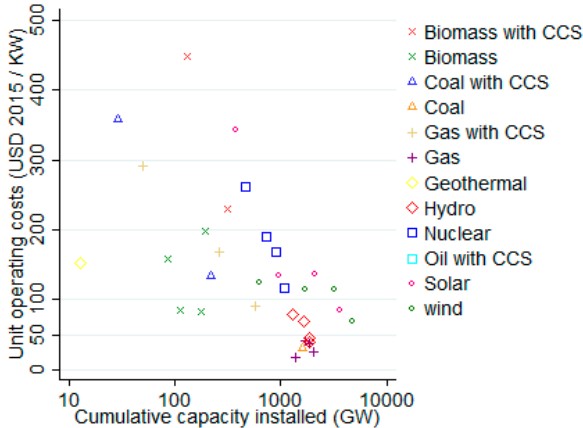

**Figure 4.** Technical cost change of power generation in the 2 °C scenario (2020–2050) (at intervals of 10 years, model mean). The representative technologies shown here include traditional thermal power generation from fossil fuels and biomass, nuclear power, and hydroelectric power. CCS represents carbon capture and storage [42].

The above results really mean that upgrading of the primary energy structure is related to higher carbon prices, whereas the proportion of clean electricity generation will increase significantly under strict climate policies.

## 3.2. Energy Investment

### 3.2.1. Energy Investment Status

According to the World Energy Investment Report 2018, the total investments of the global energy system were around USD 1.8 trillion in 2017. This amounted to about 2.23% of the GDP and 8.65% of the gross capital formation in 2017. There is no obvious change between 2017 and 2015 [19] across the total amount. As McCollum's research [19] points out, the IEA (International Energy Agency) data are basically consistent with the predicted uncertainties of the six comprehensive models mentioned in this study.

To provide a more detailed perspective on the energy investment status, Table 4 and Figure 5 disaggregate the global energy investment into regional indicators. The per capita GDP of the OECD90+EU, which are industrialized economies, is three times the global average, but the share of their energy investment in the GDP is only half the global level, and the proportion of low-carbon energy investment and energy investment in the world is similar. On the contrary, as an underdeveloped region, the per capita GDP of MAF is only 40% of the global level, but the share of the energy investment in the GDP is three times that of the global average, and the proportion of the low-carbon energy investment in the region is only 21% of the global share of energy investment.

From the perspective of the energy categories in 2015, the proportion of energy invested in fossil fuels extraction and conversion is the largest across all regions, and the investment for energy efficiency in all regions is almost zero (Figure 5). The distribution of the energy investment types in the MAF region is the most uneven, and the diversity of the energy investment types in the OECD90+EU and ASIA regions is obvious. The proportion of energy investment in fossil energy for power generation in Asia is higher than that in the OECD90+EU region, which is mainly attributed to the coal-fired power plants in China and India.

**Table 4.** Overview of regional indicators projected by six models across scenarios in 2015 (model mean).

| Region | EI Share of GDP | EI Share of Global | LCEI Share of EI | LCEI Share of Global LCEI | GDP per Capita (USD2015) |
|---|---|---|---|---|---|
| ASIA | 2% | 31% | 32% | 26% | 8181 |
|  | (1% to 3%) | (21% to 41%) | (24% to 41%) | (16% to 34%) | (7450 to 8719) |
| LAM | 2% | 7% | 34% | 6% | 13012 |
|  | (1% to 2%) | (5% to 9%) | (23% to 50%) | (5% to 8%) | (11,107 to 14,088) |
| MAF | 3% | 14% | 11% | 3% | 5532 |
|  | (1% to 5%) | (5% to 24%) | (3% to 22%) | (2% to 4%) | (5025 to 5951) |
| OECD90+EU | 1% | 27% | 34% | 25% | 39,775 |
|  | (1% to 2%) | (22% to 30%) | (28% to 42%) | (20% to 30%) | (34,735 to 46,938) |
| REF | 3% | 6% | 16% | 2% | 15,564 |
|  | (2% to 4%) | (3% to 10%) | (10% to 20%) | (1% to 4%) | (12,378 to 21,173) |
| World | 2% | 100% | 32% | 100% | 13,172 |
|  | (1.6% to 2.3%) | NC | (23% to 43%) | NC | (11,795 to 13,993) |

Note: EI indicates energy investment; LCEI indicates low carbon energy investment. The multi-model means and the ranges across the models are presented simultaneously. The values are calculated by the multi-model mean, so the numbers may not add up to the global totals.

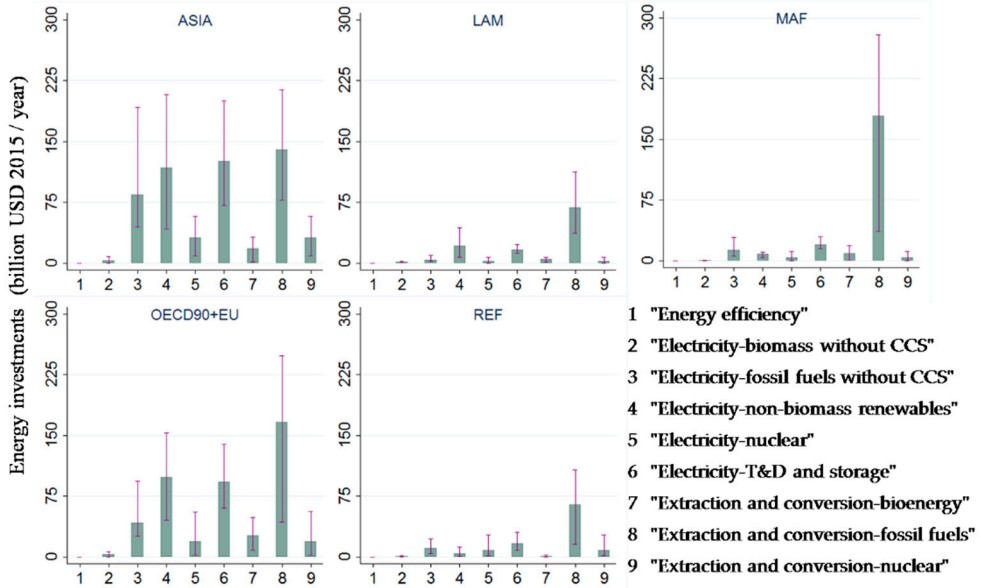

**Figure 5.** Regional energy investments by category in 2015 (billion USD 2015 per year). The bar value represents the average of multiple models; the whisker gives the min-max range of the models.

### 3.2.2. Future Energy Investment Gap Considering Carbon Price Revenue

With further increases in the population and energy demand, the overall level of global energy investment will be enhanced in the foreseeable future. Here, the energy investment gap refers to the difference between the energy investment under the climate change mitigation policy scenario (NDC, 2 °C, and 1.5 °C) and that under the CPol scenario (business as usual projection). Investment can be seen as a lever used by policy makers and investors to influence emissions [19], so quantifying the investment gap is important.

The previous literature has shown that the harsher the climate change mitigation scenariois, the wider the energy investment gap [19]. The carbon prices of all models promote investments in carbon-free technologies [44]. It is assumed that a carbon tax (carbon price revenue), ignoring administrative costs, will be fully implemented by around 2025 (based on most models), and will be an important source of energy investment. In almost all models, the annual carbon price over the period of 2025–2050 (undiscounted) is higher than in 2025, and each country introduces the same carbon tax on all GHG (Greenhouse Gas) emissions across all regions under the climate policy scenarios from 2025. There are two ways to levy a carbon tax (carbon price revenue): on either production or consumption. Here we only analyze the impact of a supply-side carbon tax on the energy investment

gap. Supply-side carbon tax is chosen here in order to be consistent with the energy investment, because the energy investment is targeted at the supply-side energy. Note that the additional energy investment mentioned here refers to the reduction of carbon tax from the energy investment, and the energy investment gap refers to the difference between the climate change mitigation policy scenario and CPol scenario. As energy investment is a tool of climate policy, it is essential to fill the investment gap for energy system transformation.

As presented in Figure 6, achieving the transformational 2 °C (1.5 °C) pathway means that the global energy investment gap in the near term (2025, 2035) will be nearly USD−864 (USD−687) billion per year, which accounts for around 52% (34%, absolute value) of all the energy investments worldwide under CPol in 2025 (2035) (model mean). Put differently, the carbon tax greatly reduces the energy investment pressure of the transformational 2 °C (1.5 °C) pathway, which means the additional energy investment needs to be minimized in 2035 (2025) for the foreseeable future. The energy investment gaps in the transformational pathways approach zero around 2040 and then changes from negative to positive. It should be noted that, if the goal is to maintain the global temperature below 2 °C or 1.5 °C for the long run, the energy investment gap will not change significantly after 2050. However, the regional results can vary considerably.

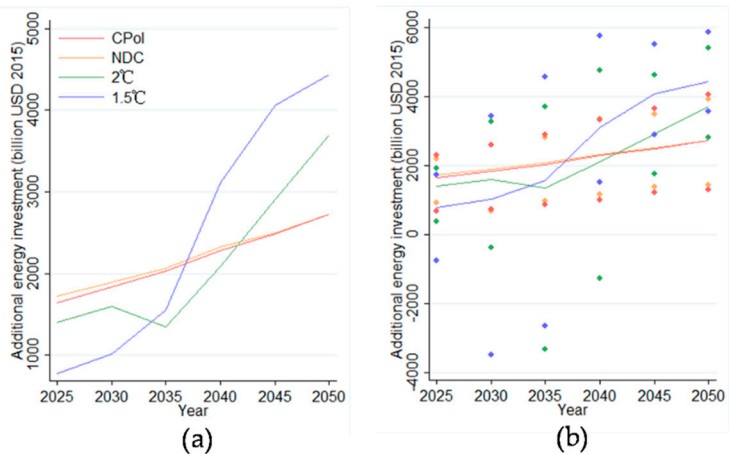

**Figure 6.** Short and mediumterm changes in global energy investment (minus the carbon tax). Note: (**a**) represents the mean value predicted by the model in different scenarios. In (**b**) extreme values of different scenarios are added on the basis of (a), represented by dots, and the colors correspond to the scenarios.

From a regional perspective (Figure 7), the energy investment gaps (model median) under the stringent pathways are positive values in Asia and OECD90+EU between 2030 and 2040, negative values in LAM and MAF, and fluctuant in REF. Interestingly, the level of energy investment (minus the corresponding carbon tax, model median) needed to achieve the NDCs is basically sufficient for the world to achieve the stringent policy targets between 2030 and 2040, which is in the LAM, MAF, and REF regions. It is also interesting to see that the near term EI-Gap for MAF is almost zero under the NDC scenario, which indicates that the increased energy investment is offset by carbon tax in the NDC scenario. These results imply that the EI-Gap will be much narrower if carbon tax is regarded as a source of energy investment.

To highlight the regional relative contribution to the energy investment gap, we added the relative value of the regional energy investment gap. Due to the large differences in the population and GDP in different regions, the relative value of the energy investment gap is expressed as the percentage of the energy investment gap relative to the energy investment in the CPol scenario (Table 5). From the perspective of this relative value, Asia contributes the most to the energy investment gap under the climate stability policies. Under the NDC scenario, Asia contributes the most in the near term and REF contributes the most in the medium term.

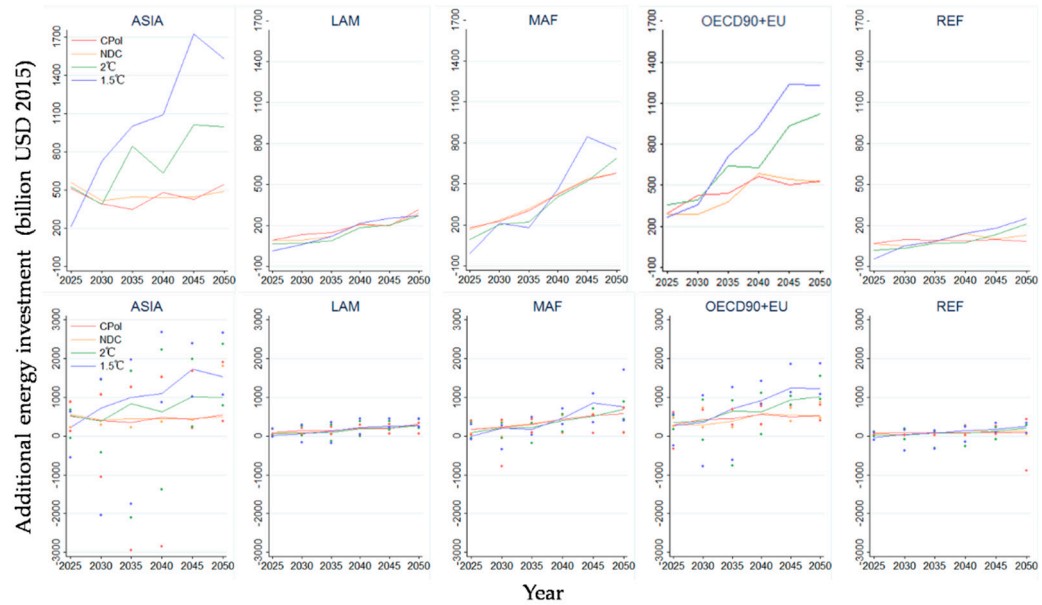

**Figure 7.** Short and medium term changes in regional energy investment (minus carbon tax). The line graph represents the median value predicted by the model in different scenarios; in the line chart with dots, extreme values of different scenarios are added on the basis of the line graph. These are represented by dots, and the colors correspond to the scenarios.

**Table 5.** The relative value of the energy investment gap in each region (model median).

| Scenario | Region | Year | | | | | |
|---|---|---|---|---|---|---|---|
| | | 2025 | 2030 | 2035 | 2040 | 2045 | 2050 |
| NDC | ASIA | 12% | 6% | 28% | −8% | 5% | −10% |
| | LAM | −3% | −32% | 6% | 2% | 2% | −8% |
| | MAF | −7% | 5% | 4% | 0% | 1% | −1% |
| | OECD90+EU | −1% | −34% | −17% | 3% | 9% | −17% |
| | REF | −2% | 11% | −18% | 65% | 68% | 34% |
| 2 °C | ASIA | 5% | −1% | 142% | 31% | 136% | 82% |
| | LAM | −33% | −49% | −23% | −12% | 4% | −14% |
| | MAF | −49% | −10% | −27% | −5% | −2% | 19% |
| | OECD90+EU | 23% | −10% | 39% | 11% | 85% | 63% |
| | REF | −69% | −33% | −24% | −13% | 115% | 123% |
| 1.5 °C | ASIA | −57% | 86% | 188% | 126% | 302% | 179% |
| | LAM | −82% | −54% | 7% | 3% | 30% | −14% |
| | MAF | −104% | −6% | −41% | 8% | 60% | 30% |
| | OECD90+EU | −9% | −17% | 54% | 61% | 146% | 96% |
| | REF | −165% | 9% | −12% | 71% | 192% | 167% |

## 3.3. Policy Costs of Different Climate Goals

Tackling global warming will certainly come with costs. The policy costs are also central to understanding and implementing the climate change mitigation policies. Here we take the policy cost as a proxy indicator to measure the difficulty of achieving the policy objectives. There may be some considerable uncertainty around the cost projections for the climate policy scenarios, such as the potential benefits of emission reductions, which are not taken into account here; this will be discussed in more detail in Section 4. This study finds that there is significant heterogeneity in the policy costs between the climate policy scenarios across time and space, but the relationship between the pledges and climate stabilization policy scenarios is robust.

This section zooms in on the share of the policy cost to the GDP (measured at market exchange rates) in different scenarios, which are presented in Figure 8; Figure 9. The policy cost of the NDC scenario in 2050 is between 0.08–0.69% of global GDP (MER), which is only about between one-tenth and one-half of the cost in the 2 °C scenario (0.94–7.87% of global GDP) and between one-twentieth and one-third of the cost in the 1.5 °C scenario (1.79–14.57% of global GDP). The global policy costs under the 2 °C (1.5 °C) scenario in all models are between 1.96% and 12.42% (3.72–20.19%) of the corresponding GDP in 2100. At this point, the policy cost of the individual model (AIM/CGE) in the NDC scenario is negative, which is not considered here. For a given condition, the higher the climate change mitigation target, the higher the policy cost and the higher the share of GDP.

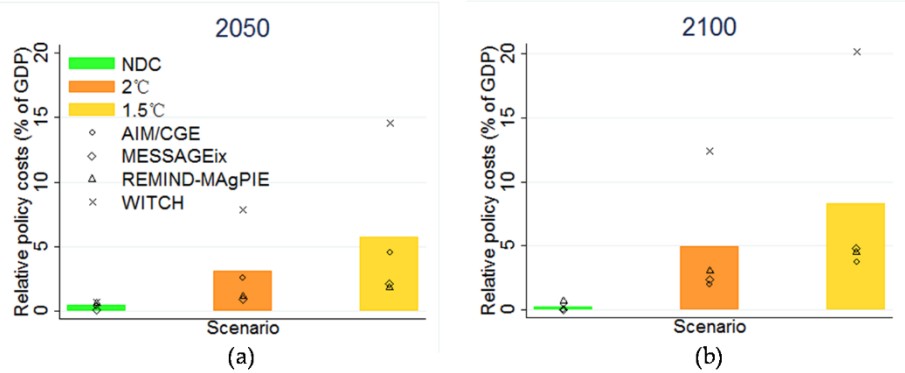

**Figure 8.** Global relative policy costs under climate policy scenarios in 2050 and 2100 (% of Gross Domestic Product (GDP)). (**a**) represents 2050; (**b**) represents 2100. The bars show the model means, and the markers show individual models. The relative policy cost is expressed by the percentage of the policy cost to the GDP (% of GDP). The colors distinguish scenarios, while the symbols represent each model studied. CPol represents "the business as usual" projection (the current situation), so there is no policy cost. Here, the GDP loss is used as an alternative indicator of the policy cost, and only four models involve this indicator.

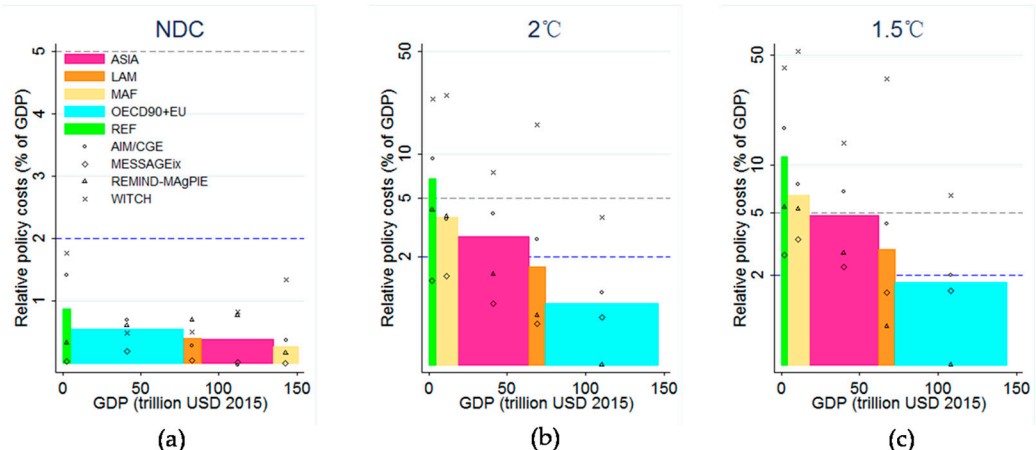

**Figure 9.** Predicted regional policy costs for climate policy scenarios in 2050 (% of GDP). (**a**) represents NDC scenario, (**b**) represents 2 °C scenario and (**c**) represents 1.5 °C scenario. Here the relative policy cost refers to the proportion of the policy cost to the GDP. The height of the bar represents the relative policy cost (model median), and the area of the bar represents the absolute value of the policy cost of each region (model median). The colors represent regions, and the symbols represent models. The Y-axis span of the two pictures on the right is too large, so for the sake of aesthetics, the Y-axis of the two pictures on the right is displayed in logarithm.

The climate policy cost as a share (median value) of the corresponding GDP in 2050 is presented in Figure 9. REF is the highest across all the policy scenarios (NDC, 2 °C, 1.5 °C), MAF is the lowest in the pledges scenario, and OECD90+EU is the lowest in the climate stabilization scenarios (2 °C, 1.5 °C). This highlights the fact that the impact of the formulation of climate policy on regional development is not uniform. Contrary to common sense, in the medium and long term, the contribution from the developed regions is the smallest in terms of the ratio of the policy costs to the economic aggregates under the climate stabilization scenarios (Figure 9).

Furthermore, Figure 10 shows the regional contribution to global policy costs under different policy scenarios, which further verifies the above conclusion from another aspect. The global policy costs of the NDC (the nationally determined contributions) scenario in 2050 are predominantly (64.5%, median value; 58.9%, mean value) borne by OECD90+EU. The policy costs of the climate stabilization scenarios mainly come from Asia and MAF in 2050, and the cost burden of OECD90+EU is less than one-fifth of the global policy costs in the climate stabilization (2 °C or 1.5 °C) scenario. It is interesting note that, in the climate stabilization scenarios, Asia's contribution to the policy costs is determined by the reduction of fossil fuel combustion and the replacement of high-cost low-carbon fuels. However, MAF's contribution is mainly accomplished by output reduction due to lower external fossil energy demand. This shows that economies dominated by the developed countries will bear more costs of climate change mitigation in the pledges scenario, while the developing and some emerging economies will bear more costs of climate change mitigation in the climate stabilization scenarios from the medium to long term.

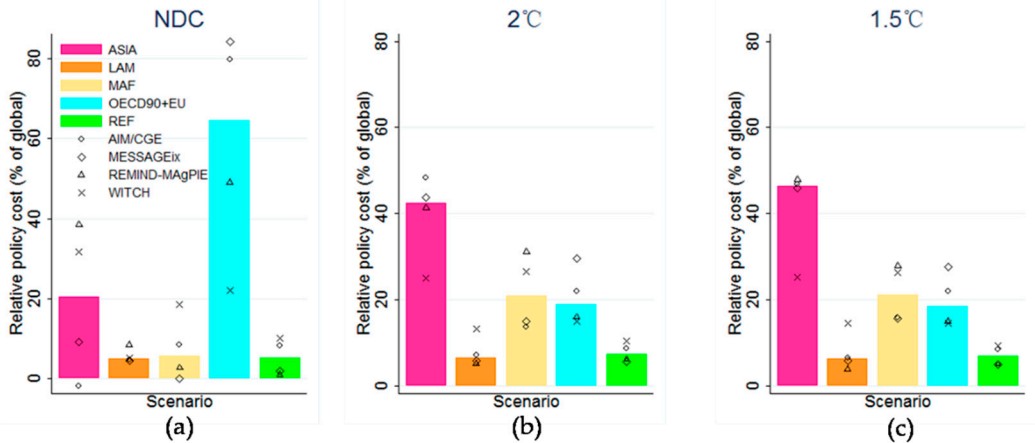

**Figure 10.** Predicted share of global policy costs by region in 2050 (model median). Note: (**a**) represents NDC scenario, (**b**) represents 2 °C scenario (**c**) represents 1.5 °C scenario. Relative policy cost refers to the proportion of the regional policy cost to the global policy cost. The height of the bars represents the relative policy cost, the colors represent the regions, and the symbols represent models.

## 4. Uncertainty Analysis

Tracking the regional contribution of climate change mitigation is by no means an exact science. We mainly use the energy and economic data related to climate change mitigation from CD-LINKS and design scenarios based on a set of assumptions. For example, most of the predictions of the NDC scenario refers to 2030 as the target year. NDC includes conditional and unconditional pledges, where the default conditional commitments are fulfilled. The scenarios of CPol and NDC do not set long term climate targets, and the relevant predictions after 2020/2030 are extrapolated based on the near term ambition levels, respectively. Meanwhile, the gap between the CD-LINKS data and the actual statistics may also affect the results.

In addition, this study assumes that carbon tax is an important source of energy investment funds, which is only a premise based on research. It is worth emphasizing that carbon tax cannot be a one-size-fits-all solution for all climate policy instruments [45]. To some extent, this will increase

the investment gap of global climate change mitigation. The results of this study can be regarded as a lower limit of the investment gap. An important shortcoming of the IAMs is that climate change will affect the rise or fall of the GDP level instead of considering the impact on the GDP growth rate. We should also note that there is considerable uncertainty about the speed of the socio-technological transitions that affect the long term emission mitigation potential.

In order to highlight the uncertainties in this analysis, we use the multi-model mean and range to report the results. Given that our analyses do not cover all possible outcomes for a given policy scenario, the means (medians) and ranges can be interpreted as consistent with SSP2 (Shared Socioeconomic Pathway2). Nevertheless, the key qualitative insights will remain the same. There may be some differences in the energy, mitigation levels, and policy actions among the countries within each group, but the uncertainties in the regional precise calculations cannot overshadow the robust indications. There is no relevant data in the IAMs at a national level; this should be explored in the future.

The fact that the United States withdrew from the Paris Accord in June 2017 is not taken into account here. To some extent, it will hinder the implementation of global climate change mitigation actions and increase the cost of the developing countries participating in global climate change mitigation. Reforming and developing economies will endure more pressure in the medium and long term as the policy scenario becomes more stringent. This view is consistent with [46].

## 5. Conclusions and Policy Implications

Our analysis provides a number of important insights, which can be summarized as follows.

1. Achieving the aim of climate change mitigation requires anenergy transition through decreasing the fossil energy supply and controlling the carbon emissions. It is necessary to set and implement strict climate policy. In the short to medium term, compared with the CPol scenario, the global primary energy supply of the NDC, 2 °C, and 1.5 °C scenarios will be decreased by 6.81% (6.90%), 19.46% (29.55%), and 26.15% (35.16%), respectively, in 2030 (2050). At the same time, under the climate change mitigation scenario, the global electricity production will decline to a certain extent in the short term (2030) and then show a growth trend.

2. Increasing the share of clean energy is the inevitable choice of energy transition, but there are different behaviors in the five regions. Under both the CPol and NDC scenarios, gas is the dominant source of electricity generation in OECD90+EU, REF, and MAF, coal is the largest power source in Asia, and hydropower is the largest power source in LAM in the medium to long term. Renewable energy is the dominant technology for power generation under the stringent scenarios (2 °C, 1.5 °C) across all regions in the medium to long run, but there are obvious differences in the specific renewable energy types between regions. Meanwhile, the capacity expansions are roughly inverse with the unit operating costs in the power generation technologies.

3. Energy investment as a tool of climate policy can be used to upgrade energy structure and reduce greenhouse gas emissions. If the carbon tax is seen as a source of energy investment, the energy investment gap will be narrowed. Under the stringent scenarios, the positive investment gap will only occur in Asia and OECD90+EU in the short term (2030–2040). This situation will not occur in LAM and MAF, while there will be some fluctuations in REF. It is noteworthy that the medium term EI-Gap for MAF is almost zero under the NDC scenario.

4. We take policy cost as a proxy indicator, which represents the degree of difficulty of achieving the policy objectives. The policy costs show significant heterogeneity in time and space, but the relationship between the pledges scenario and climate stabilization scenarios is robust. The economies dominated by developed countries will bear more costs of emission reduction in the pledges scenario, while the developing and some emerging economies will bear more costs of emission reduction in the climate stabilization scenarios from the medium to long term.

This analysis is primarily focused on the global and regional level, whereas the current distribution of contributions to climate change mitigation concerns are more apparent at the national level. IAMs

usually do not have relevant data at a national level, but it should be explored in future. Finally, to achieve a detailed understanding of the distribution of contributions to climate change mitigation, the further introduction of relevant constraints in the IAM models, will produce a more realistic description. Continuing this line of research will help to provide timely and relevant information as countries around the world take individual and collective action to address climate change.

**Supplementary Materials:** The following are available online at http://www.mdpi.com/2071-1050/11/12/3341/s1. Note S1: Five macro-regions; Figure S1: Trajectory of regional primary energy transformation (2015–2050); Figure S2: Primary energy development across models under CPol and climate policy scenarios; Figure S3: Electricity generation by technology in different policy scenarios in 2030 and 2050 at the regional level.

**Author Contributions:** The conceptualization, methodology, software, and validation, aspects were performed by H.S. and S.N.; the formal analysis, was performed by H.S. and S.N.; the data curation was performed by H.S.; the original draft as well as the review and editing, were performed by H.S.; while the visualization was performed by H.S. and X.W.; supervision and funding acquisition were performed by S.N.

**Funding:** This study was financially supported by Fundamental Research Funds for the Central Universities (Grant no. 17LZUJBWZX013). The analysis that allowed the publication of this paper was also funded by the CD-LINKS project (the European Union's Horizon 2020 research and innovation programme, grant agreement No 642147).

**Acknowledgments:** The authors are grateful to two anonymous reviewers for their valuable comments and suggestions.

**Conflicts of Interest:** The authors declare no conflict of interest.

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
