# Peer review of "Future Regional Contributions for Climate Change Mitigation: Insights from Energy Investment Gap and Policy Cost"

_sustainability, doi:10.3390/su11123341_

Reviewer 1 Report

The presented manuscript brings forth a very important a relevant subject. As such it is my opinion that it is worth publishing. However I think some of the figures need to be improved, along with some editing and english minor changes. Mainly I noticed a lot of instances where spacing was lacking between words or at the end of parenthesis. I would like to see that issue addressed. 

As for the rest of the manuscript there are some suggestions detailed in the following:

Line 20: change "study" to "studies"

Line 22: add "the" - mitigation in "the" future

Line 50-52: rewrite to become more clear. Suggestion: "Previous studies have analyzed investments in a single country or region, including Asia [18], Malaysia [19], and Latin America [20].

Line 53: Replace "including" with "include"

Lines 77-80: I don't quite see the point to this paragraph. We can see how the paper is structured as we read it. I would reccomend erasing this paragraph.

Line 85: "inconstant"? do you mean constant?

Line 92: Cut "As previous literature said", start the Sentence with "These..."

Line 131: Replace "Takes" with "Take"

Line 135: Replace "Table 2" with "Table 3"

Line 139: Replace "Table 2" with "Table 3"

Line 150: Figure 2- cannot see the values for the y axis for the EJ/year 

Line 164: Replace "maintained" with "maintain"

Line 164: I cannot understand what you mean by maturity?

Line 176: add different between "under" and "scenarios"

Line 192-193: Please re-write this sentence. I don't understand what you mean by this

Line 208: replace "datas" with "data". the word data is already plural to the word datum.

Line 210: Table 4 - you need to define what EI and LCEI are on the note of the table. It's the first time you are using this acronyms and they need to be defined. Simple put at the endo of the table EI=Energy Investment and LCEI= Low Carbon Energy Investment.

Line 224: Refer to figure 5  "almost zero (Figure 5). The..."

Line 230: Figure 5- All the graphs should be in the same scale for better comparison. This means that the values of the Y axis should be the same for all graphs in the figure.

Line 234: Replace "gaps" with "gap"

Line 237-238: Please rewrite. I would suggest something like: "Investments represent a lever that policymakers and investors can use to affect

emissions [23], and as such quantifying investment gaps is important."

Line 241-243: Please rewrite this sentence. I cannot understand if you are asking a question, if you are answering, or what is your purpose with this sentence. I think what you mean is the following - "What will happen to the energy investment gaps in NDC-achieving future transformational 2ºC and 1.5ºC pathways if carbon tax is seen as a source of investment is the topic of this section." And I would remove the paragraph, because the next sentence continues this subject.

Line 264: Can't see the values for the y axis on figure 6 (a)

Line 276: Some of the graphs are missing the numbers on the Y axis. Also the graphs should be on the same scale to allow for a more visual comparison

Line 290: Once again the graphs should have the same scale. Also there is no unit for the loss of GDP, is it percentage?, absolute value, realtive value?

Line 305: This whole figure strikes me as a bit confusing. And as always the scale of the y axis is not the same for the different graphs, which does not allow for a more visual comparison. I would advise to make this figure a bit clearer, or divide the information into two figures. It seems like there is a lot of information in each graph which makes their analysis difficult.

Line 316-317: Rewrite. I suggest - "Figure 10 furthermore shows the regional contribution to global policy costs under different policy scenarios, further verifying the above conclusion from a different aspect."

Line 329: Figure 10 needs to have the values on the y axis on the same scale for better comparison

Line 390: Replace "primary" with "primarily"

Line 441-442: Needs to be lowercase.

Author Response

Dear Reviewer 1:

Thank you for your good comments concerning our manuscript entitled“Future Regional Contribution for Climate Change Mitigation: Insights from Integrated Assessment Models”(ID: 508339). We have studied your comments carefully. Those comments are all valuable and very helpful for revising and improving our paper. According to your detailed suggestions, we have made a careful revision on the original manuscript. All revisions are clearly highlighted in the revised manuscript using the “Track Changes” function in Microsoft Word, so that changes are easily visible to the editors and reviewers. As a result of this revision, we believe our manuscript has been significantly improved.

      We thank you for constructive and helpful comments and suggestions.

Reviewer 2 Report

The document entitled "Future regional contribution for climate change mitigation: Insights from Integrated Assessment Models" proposes an analysis that is defined as a systematic quantitative analysis (multi-model with different functional structures) to answer 3 different questions. The first one is how would climate policies change energy systems be compared to a business-as-usual projection? The second is what is the distribution of energy investment gap under the premise of carbon tax as the source of energy investment? And the third is what would the regional contribution of global policy costs, especially the comparison between climate policies? This multi-model consists of applying a series of 6 different models to answer the same questions, based on a set of 4 different future scenarios and dividing the set of countries worldwide in 5 regions. The results are structured as: primary energy and electricity generation, within the effect of climate mitigation policies on energy mix; energy investment status and future energy investment gaps considering coal price revenues, within energy investment; and policy costs of different climate goals. Following is a section with some limitations of this work under the name of "Uncertainty analysis". Finally, an epigraph of conclusions is included.

In my opinion, the topic of this work is of maximum relevance, given that climate change and energy supply are two of the challenges that humanity must face in this century. However, after the meticulous review of the document I have some doubts about it. I will expose only some that I consider more relevant.

Author Response

Dear Reviewer #2,

Thank you for your comments concerning our manuscript. Those comments are all valuable and very helpful for revising and improving our paper. We hope that the correction will meet with approval.

Round  2

Reviewer 2 Report

Dear authors. I greatly appreciate the effort made to respond to all my recommendations. I insist that the subject of your work is truly relevant. I can not help but point out that I would have liked you to get away from the results, and have expressed your personal opinions about what we can all read in the newspapers. Still, I think you have done a good job.

Author Response

Dear Reviewer:

Thank you very much for your comments concerning our manuscript entitled “Future regional contribution for climate change mitigation: Insights from Energy Investment Gap and Policy Cost” (ID: 508339). Those comments are all valuable and very helpful for revising our paper. We have studied your comments carefully and have made correction. Our reply to all the comments is attached in the following pages. As a result of this revision, we believe our manuscript has been significantly improved.

We thank you for constructive and helpful comments and suggestions, and hope that the correction will meet with approval. We are looking forward to hearing from you soon.
